# Quality Assessment of Medicinal Plants via Chemometric Exploration of Quantitative NMR Data: A Review

**Abdelkrim Rebiai** [1,2,*], **Bachir Ben Seghir** [2,3,4], **Hadia Hemmami** [2,3], **Soumeia Zeghoud** [2,3,5],
**Ilham Ben Amor** [2,3], **Imane Kouadri** [2,6], **Mohammed Messaoudi** [1,7], **Ardalan Pasdaran** [8], **Gianluca Caruso** [9],
**Somesh Sharma** [10], **Maria Atanassova** [11] **and Pawel Pohl** [12,*]

1   Chemistry Department, Faculty of Exact Sciences, University of El Oued, El Oued 39000, Algeria;
    messaoudi2006@yahoo.fr
2   Renewable Energy Development Unit in Arid Zones (UDERZA), University of El Oued,
    El Oued 39000, Algeria; bbachir39@gmail.com (B.B.S.); hemmami.h@gmail.com (H.H.);
    zsoumeia@gmail.com (S.Z.); ilhambenamor97@gmail.com (I.B.A.); kouadri-2013@hotmail.fr (I.K.)
3   Department of Process Engineering and Petrochemical, Faculty of Technology, University of El Oued,
    El Oued 39000, Algeria
4   Laboratory of Industrial Analysis and Materials Engineering (LAGIM), University of 8 May 1945 Guelma,
    Guelma 24000, Algeria
5   Laboratory Valorization and Technology of Saharan Resources (VTRS), University of El Oued,
    El Oued 39000, Algeria
6   Department of Process Engineering, Faculty of Science and Technology, University of 8 May 1945 Guelma,
    Guelma 24000, Algeria
7   Nuclear Research Centre of Birine, Ain Oussera 17200, Algeria
8   Medicinal Plants Processing Research Center, Shiraz University of Medical Sciences, Shiraz 73, Iran;
    ardalan.pasdar@gmail.com
9   Department of Agricultural Sciences, University of Naples Federico II, Via Università 100, 80055 Portici, Italy;
    gcaruso@unina.it
10  School of Bioengineering and Food Technology, Shoolini University of Biotechnology and Management,
    Solan 173229, India; sharmawine@gmail.com
11  Department of General and Inorganic Chemistry, University of Chemical Technology and Metallurgy Sofia,
    8 Kliment Ohridski Blvd., 1756 Sofia, Bulgaria; atanassovamarias@uctm.edu
12  Department of Analytical Chemistry and Chemical Metallurgy, Faculty of Chemistry, University of Science
    and Technology, Wyspianskiego 27, 50-370 Wroclaw, Poland
*   Correspondence: rebiai-abdelkrim@univ-eloued.dz (A.R.); pawel.pohl@pwr.edu.pl (P.P.);
    Tel.: +213-668678553 (A.R.)

**Abstract:** Since ancient times, herbal medicines (HM) have played a vital role in worldwide healthcare systems. It is therefore critical that a thorough evaluation of the quality and control of its complicated chemical makeup be conducted, in order to ensure its efficacy and safety. The notion of HM chemical prints, which aim to acquire a full characterization of compound chemical matrices, has become one of the most persuasive techniques for HM quality evaluation during the last few decades. The link between NMR and chemometrics is discussed in this article. The chemometric latent variable technique has been shown to be extremely valuable in inductive studies of biological systems as well as in solving industrial challenges. The results of unsupervised data exploration utilizing main component analysis as well as the multivariate curve resolution, were various. On the other hand, many contemporary NMR applications in metabolomics and quality control are based on supervised regression or classification analyses.

**Keywords:** herbal medicines; NMR; chemical prints; chemometrics; biological systems

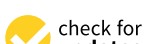

## 1. Introduction

It is difficult to undertake spectroscopic investigations of chemical mixes that contain several components [1]. These difficulties are worsened in systems with individual components that are uncertain or difficult to isolate and analyze. These systems are mixtures of

compounds in many states of order and disorder, multiphase materials, interfaces, dopants, biological systems that respond differently when isolated, and metabolomes [2]. NMR spectroscopy is a good tool for examining these complicated mixtures since it can quantitatively evaluate the specification of a bulk sample while also revealing information regarding the atomic environment, the ordering within the sample, and the electronic structure [3]. Even the statistically based algorithms, such as principal component analysis and free component examination, can also be used to separate spectra components into successive sections without having comprehensive knowledge of the source signals associated with these components. These techniques are useful tools; however, due to implementation concerns, such as software challenges, a lack of understanding, and a lack of literature examples, they are not commonly employed.

The present article discusses many of the most frequently utilized latent variable approaches in NMR data analysis and predictive modeling. It describes how difficult it is to transform NMR data for multivariate data processing, and how various chemometric techniques affect NMR data.

## 2. Herbal Extracts as Therapeutic Agents

For millennia, traditional herbal therapy has been used in a variety of ways, and it is still mostly used as a primary form of healthcare in a number of developing and disadvantaged nations [4]. Traditional medicines (TMs) were used by nearly 80% of the world's population in impoverished nations for basic healthcare due to their cheap cost and accessibility. According to the WHO, aborigines residing in rural villages are not served by contemporary treatment facilities [5]. Early explorers' plant collections, as well as ethnobotany, have played a significant role in the discovery of novel pharmaceuticals for many centuries. Plants and their derivatives account for 25% of all medications in developed countries [6]. In many African and Asian nations, medicinal plants are still used in basic healthcare by up to 80% of the population [7]. These herbal medicines (HMs) are also gaining prominence in developed countries, particularly the United States and Germany.

Despite its existence and continued use over many centuries, traditional medicine has not been officially recognized in most countries. Therefore, due attention and support has been paid to education, training, and research in this area. The quantity and quality of the safety and efficacy data on traditional medicine seem to be insufficient to meet the criteria needed to support its use worldwide. The reasons for the lack of research data are primarily healthcare policies as well as the lack of adequate or accepted research methodology for analyzing TMs [8]. To analyze the quality and authenticity of HMs, the identification of specific herbs and their main components is essential. However, this examination does not provide the whole picture for HMs because multiple factors are often responsible for their therapeutic advantages. The majority of phytochemical constituents of herbal products must be established in order to ensure the reproducibility and reliability of pharmacological and clinical research on these products, to better understand their bioactivities and potential side effects, and to improve product quality control [9].

It is crucial that we identify the variation between conventional medicines and HMs when comparing these two types of therapeutic agent and how they are supplied. The administration of a pure chemical in comparison to a plant extract containing the same chemical entity is quite different. The distinction is mostly due to the intricacy of a plant extract, which introduces a flood of variables into conventional phytomedicinal research, all of which could influence its chemical complexity and bioactivity. Weathers et al. (2011) found that when a plant sample (e.g., *Artemisia annua*) was administered vs. a pure medicine (e.g., artemisinin), the bioavailability of the bioactive substance through the leaves was 45 times higher than in the case of the pure drug [10]. As a result, the plant extract's complexity may have contributed to the higher bioavailability of the bioactive substances, and consequently their bioactivity.

Herbal therapy may be critical since science is only beginning to appreciate the numerous complex, diverse, and sophisticated mechanisms that operate in a wide variety of biochemical systems found in organisms [11].

HMs have shown a wide range of success in treating infections, particularly in recurring and chronic illnesses. Additionally, it is worth noting that a variety of plant extracts, having a variety of bioactive compounds, can be used to achieve clinical efficacies that are typically not possible with single-compound-based drugs, not to mention to provide critical combination therapies that affect several pharmacological targets [12,13].

### 3. Quality Control and Quality Assurance (QC/QA) of Herbal Medicine

The presence of active components with medicinal benefits in HMs depends on various factors, such as the period and place of harvest of the medicinal plants, the type of soil in which they are planted, the quality of water used for their irrigation, and the way the HMs are prepared [14]. HMs and the included treatments are part of a broader field of complementary and alternative medicine [15]. The quality control of traditional medicines is a crucial problem attracting a lot of research attention, since the safety and efficiency of TMs are intimately linked to their quality [16]. Over the last few decades, the quality requirements for HMs, herbal drug (HD) preparations, and herbal therapeutic items have advanced dramatically [17].

High-performance liquid chromatography with diode array detection (HPLC-DAD), liquid chromatography with mass spectrometry detection (LC-MS/MS), and gas chromatography with mass spectrometry detection (GC-MS) are the commonly reported methods for detecting unlawful adulterations in HMs [18–21]. Recently, the local straight line screening (LSLS) technique was devised for resolving complicated IR spectra of potentially contaminated HMs [22]. Chromatographic and electrophoretic methods in combination with various detectors, such as IR-LSLS, and nuclear magnetic resonance (NMR) [23–25], have been documented in relation to the adulteration of herbal formulations advertised for weight loss [25–28].

All parts that contribute to the superiority of HDs should be considered in standardization methods, including sample identity, organoleptic assessment, pharmacognostic assessment, volatile matter assessment, quantitative assessment (ash values, extractive values), xenobiotics assessment, microbial load assessment, phytochemical assessment, toxicity assessment, and biological activity assessment [29]. Researchers have used chromatographic fingerprinting methods to assess the quality of herbal samples and the products developed from them. To protect the safety of the consumer, sample identification must be conducted with extreme caution. It is accomplished by removing adulterations (plants mixed together) or full misidentifications (wrong plants), as well as samples of low quality (low quantities of active chemicals) or possessing excessive concentrations of pollutants (e.g., pesticides) [30]. Chemometric techniques are now being utilized in conjunction with chromatographic data to generate even more accurate data for determining the integrity of HMs as well as for observations on the comparisons and differences between HMs data. The strength of chemometrics is in the multidimensional remarks that are used to describe the data's similarities and contrasts, which are then presented in a graphical manner that is user-friendly [7].

### 4. Techniques in Metabolomics

Metabolomics is the comprehensive quantitative and qualitative analysis of all metabolites existing in biofluids, cells, tissues, or organisms. Metabolites are the end products of biological processes [31,32]. The orderly study of small-molecule metabolites, which are byproducts of certain biological activities, is known as metabolomics [33].

Due to the formidable complexity of biological systems, particularly those of plants, one-step analysis and visualization of all metabolites in a metabolome is not practical, in contrast to other "omics" approaches, such as genomics, proteomics, and transcriptomics. The four major areas of metabolic analysis are as follows:

(a) The quantification of specific metabolites via targeted compound analysis.
(b) Metabolic profiling for determining the quantitative and qualitative characteristics of a set of related substances or specific metabolic pathways.
(c) Metabolomic fingerprinting for classifying samples via quick global examination.
(d) Metabolomic study that entails analyzing "all" metabolites quantitatively and qualitatively (which is not yet possible).

Targeted metabolite analysis, also known as metabolite profiling, examines a subset of metabolites in samples using a specific set of analytical techniques, for example, GC-MS and LC-MS, to estimate their quantity, rather than perform a complete metabolome analysis. Thin-layer chromatography (TLC), Fourier transform infrared spectroscopy (FT-IR), and NMR and Raman spectroscopy are just some of the other methods employed in metabolite analysis [34,35]. Recent advancement in analytical techniques used for detecting and characterizing low-mass compounds, e.g., MS and high-field NMR, has resulted in a particularly fast method for analyzing metabolite data matrices generated by metabolomic investigations [7]. Due to the rapidity and robustness of NMR, which is crucial for industrial quality control, this study focuses on its use.

As a consequence of the relatively high sensitivity and extensive presence of protons in organic metabolites, NMR spectra provide a rich source of information on the content of metabolites in samples. NMR spectra can be generated in 10 min using 10–50 mg of material. It is usually possible to identify 10–20 recognized chemicals based on an NMR spectrum that may comprise even 50–100 metabolites [36]. NMR spectra are acquired in deuterated liquids using an NMR spectrometer (ideally 400 MHz or higher) set to the appropriate proton NMR frequency of the instrument. A number of scans are performed on each sample, ranging from 64 to 256 for high-quality spectra. The number of necessary scans is determined by the NMR spectrometer's magnet strength; as magnet strength increases, fewer scans are required. The (1) relaxation delay and (2) pulse width parameters are two important factors to be considered when obtaining high-quality spectra [37]. The concept of using all data points in an NMR spectrum is gaining traction, and for some sophisticated algorithms that align the peaks and eliminate any undesired variances, it will become a far more prevalent practice [38].

## 5. Plant (Herbal) Metabolomics Experimental Methods

### 5.1. Herbal Product Collection and Extraction

Traditional collection of plant material must consider a number of aspects that might have a substantial impact on the sample's integrity (e.g., collection time, weather, season, soil, etc.) [7]. The processing of herbal material is distinct from that of normal metabolomics analysis in that the HM material is treated before processing, and the samples are taken into the production line as a processed herbal product. It is nevertheless critical to obtain an accurate representation of the herbal plant material that will be used to make the herbal product. To avoid introducing any undesirable variance in the data, good manufacturing procedures (GMP) and good laboratory practices (GLP) must be followed.

The extraction method may cause metabolic events to occur in the plant material, resulting in a change in the sample's metabolome. As a result, it is critical to consider the chosen HM preparation procedure. The analytical sample must be prepared in the same way that the HM is prepared for consumption by patients. To remove the metabolites from the cells, the HM product must be crushed and extracted; this is performed at low temperatures and/or in the presence of a solvent. Ultrasonication has been demonstrated to be the most efficient method for degrading plant cells, and thus for producing the highest quantity and diversity of the metabolites for study [39].

It is vital to note that the extraction technique must account for both the solute–solvent interactions and the solute or analyte's dissociation from the matrix. This means that, in addition to a proper solvent choice, the techniques of pre-extraction and matrix treatment during extraction play a significant role in metabolite release, and that, in order to achieve good results, not only the type of solvent used, but also the physicochemical

characteristics of the matrix, the effect of pH on the matrix, the contact time, and the metabolite compartmentalization, should be taken into account [40].

NMR is one of the possible analytical techniques employed in metabolomics that offers significant benefits over other techniques. This is not only because of the great information richness of the resultant spectra for chemical constructions, it is also due to the good repeatability of NMR chemical changes, the simple comparison of relative metabolite amounts, and the fewer pre-separation processes required before analysis [41].

### 5.2. NMR Analysis

[1]H NMR spectroscopy is a method that can provide a "metabolic fingerprint", which can be used to determine the sample's overall biochemical makeup. It is possible to monitor the changes in the concentrations of thousands of metabolites instantaneously by comparing the spectra from several samples, and so to observe the dynamic metabolic profiles [42]. This principle (also known as "metabonomics", "metabolomics", "metabolic fingerprinting", and "profiling") has been used in a variety of fields, including the quantification of drug toxicity [43,44], environmental metabolism [45,46], plant metabolism [47,48], and, more recently, natural product profiling [49]. Chemometric data analysis techniques can be used to develop mathematical design models to predict structure, activity, and metabolic relationships based on NMR spectroscopic data (or indeed any other multivariate analytical data) (Figure 1) [41].

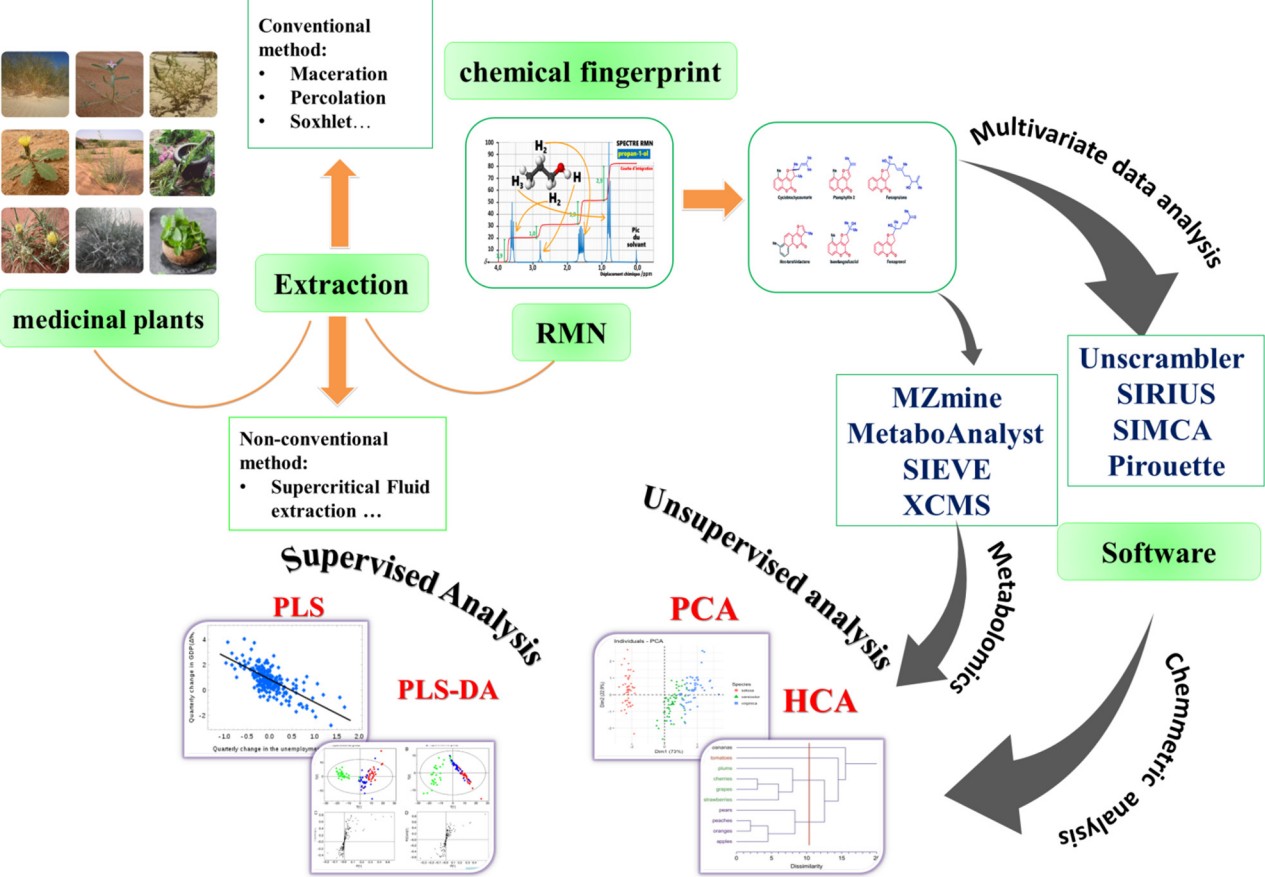

**Figure 1.** Flowchart of the investigation procedures involved in the examination of natural products.

In NMR metabolomics, significant volumes of plant material are not required. In 10 min, 10–50 mg of plant material is enough to obtain its [1]H NMR spectrum. It is generally easy to identify 10–20 recognized substances from an NMR spectrum that may comprise 50–100 metabolites [50].

## 6. Chemometric Analysis

One-to-one correlations between a cause and its effect are frequently required in science in order to validate relationships; this can be accomplished, usually, via modest data investigation in conjunction with a randomized experimental scheme [51]. Although univariate causality definitely has a purpose, it has become a barrier to understanding more complex systems, such as those seen in metabolomics, which conceals many relationships. To identify these relationships, multivariate techniques for data exploration are necessary [50].

The majority chemometric methods work with PCs (principal components) from the data's underlying common or latent patterns. Among its principal advantages for data processing, the most important is its capacity to manage large multivariate datasets using variable collinearity and projecting multivariate data into a few dimensions that can be shown visually. The disadvantage of using PC models is that they demand the simultaneous analysis of data from several samples, which might lead to the loss of qualitative information [51]. All these data must be low-rank and bilinear for the successful application of chemometrics to spectral data. In reality, bilinearity means that the signal intensities in the example spectra need to be proportionate and additive to the concentrations [52]. Additionally, a chemical component must have a single spectral signature (for example, a collection of multiple spectra); thus, it must have the same spectral shape across all samples, with the exception of intensity [53].

### 6.1. Unsupervised Data Exploration by PCA

The most basic and often-used unsupervised chemometric approach is principal component analysis (PCA). PCA has been around for more than a century [54]. Owing to its extremely powerful data reduction and data display characteristics, it might be called the parent approach for exploratory multivariate data analysis.

PCA is a statistical test that belongs to the factor analysis category. PCA is a mathematical tool that uses a small number of factors to represent the variation in a dataset (i.e., responses used to characterize the samples). A two-dimensional or three-dimensional projection of samples is commonly produced for visual analysis, using the axes (principal components, PC) as factors. Each PC is a linear combination of the original responses (with some correlation between them), and they are orthogonal to one another [55]. The number of PCs used, and hence the amount of variance collected, should be carefully chosen. One method is to count the number of components that produce the best classification model, or to use cross-validation to identify the PCs. Another factor to consider is how the score plot is visualized. The PCs should capture a high fraction of the variation if one is interested in observing the natural aggrupation of the data. Low-percentage PCs do not explain raw data well. Another difficulty is the requirement for data pre-processing before PCA application. The chosen pre-processing method is determined by the nature of the data, and should be explored [56]. Three issues were raised in a recent review [57]; i.e., (1) incorrect units for exploratory analysis, (2) misinterpretation of PCA loadings from a first- and second-derivative pre-processed signal, (3) artefacts caused by signal normalization procedures, such as SNV or MSC, leading to the misinterpretation of PCA loadings. As a remedy to these issues, the units employed should be directly connected to the concentration to minimize misinterpretations. According to the standard PCA rule, a sample with a high score value at one PC is defined by the variables that are positively linked with the high loading values for that PC. This is not always the case for the derivative spectra. The interpretation of PCA loadings based on the first or second derivative is not straightforward, and it is necessary to evaluate the derivative signal as well as the loadings. The application of an anti-derivative function on PCA loadings is recommended as a solution in this case. Normalization can remove not just the undesired signal variations, but also the important information from the original spectra [57].

PCA separates the data matrix, such as a sequence of the NMR spectra, into PCs, which are linear combinations that, in a least squares sense, approximate the original

dataset [58]. The first PC in the data is the spectral profile (loading) that best describes the largest proportion of the variation; the second PC is the profile that best describes the second largest, and so on. The PCs are composed of what is known as scores and loadings. The loadings possess the information on the spectral variables, whereas the scores contain the information on the loading vector's amount or significance (pseudo-concentration) in the sample set (chemical shifts). The residuals are the parts of the data that the model does not explain (E). The new variables are uncorrelated and are formed by combining the new coordinate system based on the directions with the greatest deviations (the first PCs). PCA achieves this by efficiently deleting a small number of orthogonal PCs, making it the most robust and consistent chemometric method available [59,60].

### 6.2. Self-Modeling Multivariate Curve Resolution

PCA is typically used for the categorization and exploration, and because the loadings and scores must be orthogonal, it cannot directly estimate real chemical spectra and concentrations. However, an unsupervised classification approach that does not require orthogonality can also be applied; for example, in the case of the multivariate curve resolution (MCR) [61,62]. In the NMR literature, this is referred to as molecular factor analysis (MFA) [63,64].

### 6.3. Supervised Data Exploration by PLS Regression

PLS (partial least squares) is a frequently used method for modeling the relationships between various sets of observed variables using latent variables. Wold and coworkers created PLS projections of observed data regarding its latent structure [65]. The PLS method incorporates regression and dimension reduction techniques, as well as modeling tools to modify the relationships between sets of observed variables by a small number of latent variables (not directly observed or measured). These latent vectors, in general, maximize the covariance between several sets of data. Similar to PCA, it can be used as a discrimination tool and a dimension reduction method [8,9]. It is also related to other regression techniques, including principal component regression (PCR), ridge regression (RR), and multiple linear regression (MLR); all of these techniques can be grouped together under the umbrella of continuum regression (CR) [66].

The counterpart of regression analysis is partial least squares (PLS) regression [63], which is fundamentally based on the PCA principle. On the other hand, the latent variables of spectral data are identified in such a way that only the information needed to forecast the physical/chemical data is recovered. PLS regression aims to develop a linear regression model that can predict a desired attribute from a multivariate signal. While PCA can be compared to going shopping for data without a shopping list (i.e., the data analysis is performed without any prior knowledge), PLS regression can be compared to going shopping with a specific shopping list (in the data). The calibration models between the NMR spectra and a specific response variable can be created if the reference data are available. This can be accomplished using PLS regression, which is the second most fundamental technique in chemometrics. The bilinear data matrix (X) is resolved into linear components by PLS, similar to PCA (latent variables). PLS, on the other hand, focuses on data variance that covaries (or is related when auto-scaled) to the response variable(s) (y). As a result, PLS regression is a supervised approach for developing prediction models that can be used to substitute a slower, more precise, and accurate analytical method with a faster, more precise, and accurate NMR-based method. PLS is a powerful regression technique that specializes in identifying and fitting erroneous variables with random correlations to a reference value. To minimize overfitting in highly empirical chemometric models, an adequate validation technique must be used (fitting the noise). The ideal condition is to have a distinct dataset against which to evaluate the developed models' genuine prediction performance. As this is not always practicable or practical, cross-validation is used instead [65,67], and can be useful for achieving the robust results (Figure 2).

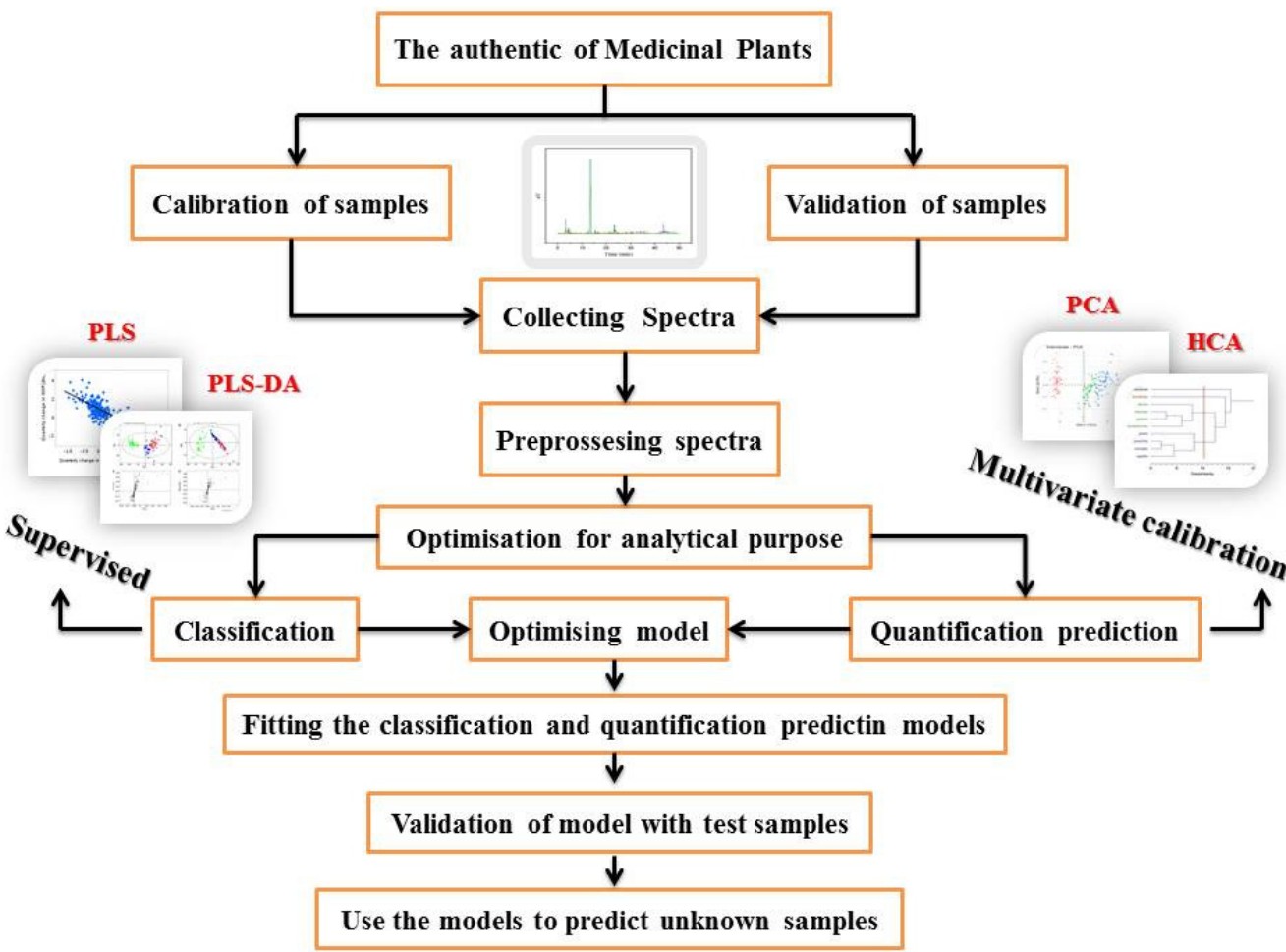

**Figure 2.** The general analytical workflow of chemometric methods.

Cross-validation divides the sample set into a number of segments, each of which is excluded, one at a time, and used as a pseudo test set for a model constructed from the remaining samples in order to estimate the prediction error, i.e., the root-mean-square error of cross-validation, and thus, to determine the number of latent variables (components) in a PLS model (RMSECV) [68]. The model data demonstrate how the RMSECV changes as the number of components increases. The initial minimum of the RMSECV curve is chosen as the appropriate number of components for the model. In addition to the error measure, model performance is typically presented as a projected vs. measured plot (RMSECV).

### 6.4. Supervised Classification by PLS-DA Regression

Despite the fact that PCA is a powerful exploratory tool, researchers are typically dissatisfied with the discrimination provided by unsupervised PCA; hence, supervised classification approaches are used instead. When creating a classification model using supervised classification chemometric approaches, a priori knowledge is actively utilized. The partial least squares regression discriminant analysis (PLS-DA) approach is the archetype of the supervised classification methods [69]. PLS-DA is highly similar to a normal PLS regression model [65], in which the major goal of PLS-DA is to discriminate between two or more classes into which the data can be classified, rather than predicting a continuous outcome.

The PLS-DA method achieves this discrimination by utilizing a pseudo response variable (typically a dummy vector with zeros and ones describing the subjects' affiliation with one of two classes), and attempting to predict it as accurately as possible using the information from the metabolite table or the NMR spectral dataset. The objective of this classification exercise is to identify a set of latent metabolite patterns (components) that most

accurately classify two (or more) groups. Classification parameters are used to optimize the PLS-DA model (e.g., rate or percentage of misclassified samples). Due to their high classification performance, PLS-DA models are widely used in NMR metabolomic research.

### 6.5. Improved Supervised Models by Variable Reduction

Despite the outstanding performance of PLS and PLS-DA, the pursuit of superior regression/classification performance involves the employment of more complex algorithms in the experimental design of bioactive chemical therapies to harness any conceivable relevance. Orthogonalization and variable selection are the two most frequently used approaches.

NMR spectra frequently reveal the regions devoid of chemical information, which can disrupt the subsequent model, at best, and obscure any useful information in misleading correlations, at worst. When the most informative spectral information is concentrated at a few extremely small peaks, or even a single peak, against a background of numerous larger peaks that vary in ways unrelated to the groups of interest, neither PCA nor PLS is sensitive enough [70]. Variable selection appears to be an obvious strategy to improve the regression or classification performance in such cases. As NMR studies involve such a large number of variables, many of them will be meaningless (irrational, noisy, or signals conveying information not relevant to the study); deleting these variables will often lead to better classification models.

There are numerous ways of eliminating variables in order to improve the performance of classification or regression algorithms, as well as data interpretation. Interval PLS (iPLS), a fundamental technique for variable selection, has been shown to be particularly useful for NMR spectra in terms of improving and simplifying regression or classification models based on NMR data by subdividing the variable space (parts per million scale) into smaller intervals [71–75]. iPLS is a PLS extension that creates local PLS models for a variety of subintervals across the complete spectrum. The fundamental benefit of iPLS is that it delivers an overall view of the vital information in several spectral subdivisions, reducing the interference from other areas [76].

iPLS identifies spectral zones that contain information about the response variable (y). The goal of this method is to limit the amount of possible interferences by reducing the variable space (intervals can be as little as one variable in size), resulting in a more compact and often superior model. Interval models, which divide up distinct portions of the NMR spectrum into logical and more homogeneous sectors, such as aromatic, carbohydrate, and aliphatic sectors, are generally a good component of NMR fingerprinting.

### 6.6. Improved Supervised Classification by Orthogonal Factor Extraction

Gender, age, and time of year are just some of the elements that cause diversity in biological systems. R.A. Fisher (the founder of modern statistics) recognized this, and developed the experimental designs that could estimate and handle variation caused by these factors. Models for analyzing univariate data from intended experiments include the paired t-test and analysis of variance (ANOVA). These methods are based on estimating the variance associated with a nuisance (orthogonal) factor (e.g., subject) and removing it. This emphasizes the variety of interest, such as treatment, which enhances the chances of discovering anything intriguing (often referred to as statistical power). Multilevel PLS-DA is a multivariate equivalent of two algorithms [74]: analysis of variance–concurrent component analysis (ASCA), and external parameter orthogonalization (EPO) [77–81]. This is an outside factor. The essence of ASCA, multilevel PLS-DA, and EPO is the partitioning of data into two (or maybe more) orthogonal subspaces, one of which is disconnected, and hence irrelevant for the study design component under consideration, and the other contains the relevant data. Whereas variable selection can be thought of as a horizontal interference elimination, orthogonalization can be thought of as vertical interference elimination in the data matrix. OPLS-DA is a widely used approach in metabolomics that includes orthogo-

nalization [78]. OPLS-DA divides information into two orthogonal subspaces, similar to multilevel PLS-DA, ASCA, and EPO.

In contrast, OPLS-DA does not use any data other than the factor of interest (e.g., treatment). Obviously, OPLS-DA model prediction/classification performance is identical to that of the corresponding PLS-DA model, and OPLS-DA's popularity stems solely from its easier and more obvious interpretation [82].

## 7. Multivariate Data Analysis

NMR has a distinct advantage over mass spectroscopic techniques in screening and fingerprinting applications since it consistently identifies most of the chemicals present. Mass spectroscopic techniques are plagued by issues caused by the varying ionization of various types of compounds [83]. Multivariate statistical approaches, such as PCA, are commonly used to analyze NMR data initially. PCA is a data visualization technique that may be used to spot patterns in large datasets. PCA and other similar multivariate analyses may be performed using a variety of commercially available software tools [84]. PCA is the statistical analysis technique for extracting and visualizing the systematic variation in data [85].

PLS, like PCA, may be used as a discriminating tool and a dimension reduction approach [86]. However, there appears to be an issue with dealing with vast amounts of data, requiring the use of multivariate statistical data analysis (MVA), also known as chemometrics, to recover the quality attribute information buried by screening approaches. MVA developments, such as PCA, linear discriminant linear analysis (LDA), soft independent modeling of class analogy (SIMCA), and PLS regression, and qualitative techniques, such as the cluster analysis (CA), are used in combination with various pre-processing methods to extract the required information from convoluted spectra [87].

To obtain values for the PCA score plot, the original variable from the NMR/MS data is multiplied by coefficients, sometimes referred to as loadings. The exact numerical value of the loading will indicate the link between the original variable and the component [7]. As a result, "loading plots" may be used to emphasize the spectral zones accountable for data separation and, as a result, for the precise location of the scoring plots [88].

### 7.1. Adaptation of NMR Data for Chemometrics

The Fourier transformation of the time-domain (seconds) (FID) of NMR signals recorded as a function of time can be used to derive the chemical shift spectrum (FID-free induction decay). The quality of the spectra, and thus the quantitative information contained within them, are determined not only by the strength and homogeneity of the external magnetic field (shimming), but also by the precise and accurate tuning and matching of the nuclear magnetic resonance frequency [89], phasing, baseline correction, line shape correction, and chemical shift referencing, as well as pH, temperature, and water filtration [90]. When applied to large NMR datasets, chemometrics or any other quantitative statistical method puts the analytical NMR platform to the test in terms of the long-term stability parameter.

The fact that the observed resonance frequencies are extremely sensitive to the local chemical environment is probably the most valuable feature of $^1$H NMR spectroscopy, but this shift in sensitivity also means that the resonance frequencies can be affected by some minor temperature, pH, and external magnetic field fluctuations [91]. To solve this problem, a variety of peak (signal) alignment algorithms has been developed. While the overall spectrum-to-spectrum variations caused by some small changes in the spectrometer frequency can be resolved by a simple translation of the entire spectrum, using either a pattern recognition method or an internal reference peak, the local peak-to-peak chemical shift variations caused by, for example, pH and temperature, are more challenging to handle. Smoothing and/or data reduction via binning are two viable solutions to the problem (bucketing) [92], but several more generic and elegant alternatives, such as correlation optimal warping (COW), have been also devised [93], in addition to recursive segment-

wise peak alignment [94] and interval correlation optimized shifting (icoshift) [95]. When processing the multivariate data from large-scale NMR metabolomics experiments, there are often two alternatives, which are outlined below.

### 7.2. Spectral Fingerprinting

In this case, multivariate data analysis makes use of all of the spectral properties. The goal of this technique is to compare the "fingerprints", or patterns of changes in response to bioactive molecule intervention, rather than to identify each observable metabolite. As processing capacity has increased and metabolic fingerprinting can reveal some novel and unexpected insights, there is a growing trend to undertake the analysis at the highest spectral resolution available [96]. Metabolic fingerprinting produces a dataset with the following dimensions: number of samples and number of spectral variables, where the number of spectral variables is typically (many) hundreds.

### 7.3. Spectral Profiling

For each sample, as many signals as are conceivable are discovered (although not necessarily assigned), and their peaks are combined. This method will provide a dataset with the following dimensions: number of samples and number of peaks, where the number of peaks/metabolites is frequently in the hundreds. Spectral profiling complicates pre-processing by requiring integration settings, but it produces more comprehensive data tables for further data analysis [97].

## 8. Natural Product Metabolomics Using NMR

### 8.1. Analysis of Complex Pharmaceutical Preparation Using NMR Metabolomics

HMs are often quite complicated and contain a large number of chemicals in the commercial market. Their preparation is frequently standardized using single-indicator compounds or a set of connected compounds that do not contain the information on other, apparently irrelevant, copious ingredients present in the herbal mixture. NMR metabolomics also reveal the significant fluctuations in the concentration of flavonoids connected to the *Hypericum perforatum* extract anti-depressant effect. Full-resolution NMR data yielded plots with more accurate information on the chemicals in the extract responsible for the grouping, and possibly for the therapeutic effect, showing that full-resolution NMR data may be preferable for PCA of plant extracts and HMs (Table 1) [98].

### 8.2. Using NMR Metabolomics to Analyze Complex Artemisia Herbal Medicine

The combination of NMR spectroscopy with PCA has proven to be a highly promising approach for the detection of specific ingredients in herbal extracts that were stated as present throughout the extraction process. The presence of active anti-malarial artemisinin in capsules manufactured from *A. afra* was studied [97]. NMR analysis was performed on *A. afra* extracts and *A. annua* extracts, and the product capsules. The analyses were carried out using a 500 MHz Bruker NMR spectrometer, with 128 scans completed for each sample. At 6 min per sample, the spectra were referred to the residual chloroform peak. For PCA, NMR data were processed and files were transmitted to SIMCA-P (10.0 Umetrics, Umea, Sweden).

The differences between the three samples were clearly visible in this analysis. The plant species *A. afra* and *A. annua* can be readily distinguished from one another using PCA data, separating extremely well in the first PC. The capsules were grouped together with *A. afra* in PC1, and separated from *A. annua*, indicating that the capsules certainly contained *A. afra*. The *A. afra* and capsule samples were well-separated in PC2, with the difference in chemical concentration between them explaining the separation. The anti-malarial component, artemisinin, was only discovered in the *A. annua* samples, and not at all in the *A. afra* samples, according to the same study's LC-MS results. Thus, utilizing NMR-based metabolomics, the stated presence of the anti-malarial component, artemisinin, in the capsules of *A. afra* was revealed to be false [99].

### 8.3. Chemical Profiling of HMs

The majority of existing standard QC procedures for entire extracts are insufficient for use on HMs. There are a lot of differences between the batches of items on the market. As many factors can affect the chemical composition of phytomedicine (plant growing environment, collection/harvesting season, preparation and extraction procedure, and so on), it is vital to guarantee that these elements are considered during HM quality control [100]. High-resolution [1]H NMR may be quite useful, and when paired with chemometric analysis, it can be used to evaluate the entire plant extract. It allows one to see "all" of the chemical components in a plant extract at the same time, as its "metabolic fingerprint". The differences and similarities between the economically relevant samples may be readily and rapidly seen in 2D or 3D plots using PCA algorithms on NMR data, adding to the high-throughput demand for QC procedures [46].

### 8.4. Ephedra Metabolic Fingerprinting Using NMR

The *Ephedra* genus is one of humanity's earliest therapeutic plants. *Ephedra sinica* is the most common source of ephedrine alkaloids, although ephedrine alkaloids can also be found in the 14 species mentioned in this subsection. As a result, the need for species identification is critical, and the present study demonstrates that this may be accomplished using NMR-based metabolomics. NMR was used to examine three *Ephedra* species (*E. sinica*, *Ephedra intermedia*, and *Ephedra distachya* var. distachya) and to compare them to nine commercially available *Ephedra* herbal plant samples acquired from the Taiwanese market. In PC1, the three known *Ephedra* species were separated quite effectively, with *E. distachya* var. *distachya* separating totally from the other two species, which was confirmed to be owing to the absence of ephedrine. The aqueous extracts were separated most strongly; thus PCA was performed exclusively on them. Except for one species that was grouped between *E. sinica* and *E. intermedia* and was proven to be a hybrid of these two, the majority of commercial samples were clustered close to *E. intermedia* (Table 1).

It is obvious that NMR-based metabolomics may be utilized to assess the authenticity of herbal plant material quickly and simply. Due to the fact that a considerable portion of the globe still relies on traditional herbal markets to receive their primary treatment, authenticating these supplies of HMs can be quite beneficial. Some HM firms rely on wild-gathered plant material for their herbal manufacture, thus it is vital to double-check the materials before employing them in the production process [101].

**Table 1.** Applications of NMR in chemometric analysis of natural products.

| No. | Plant Name | Publication Year | Research Aims | Applied Statistical Methods (If Any) | Analytical Techniques | References |
|-----|-----------|-----------------|---------------|--------------------------------------|----------------------|-----------|
| 1 | *Panax ginseng* (roots) | 2012 | Quality assessment | PCA<br>HCA<br>PLS<br>PLS-DA | [1]H-NMR | [102] |
| 2 | *Artemisia afra*, A. annua (herb) | 2010 | Quality assessment | PCA | [1]H-NMR | [103] |
| 3 | Zea mays (seeds) | 2010 | Quality assessment | PCA<br>HCA<br>SIMCA | [1]H-NMR | [104] |
| 4 | *Echinacea purpurea, E. pallida, E. angustifolia, E. simulate* (roots and aerial parts) | 2010 | Taxonomic discrimination | PCA | [1]H-NMR | [105] |
| 5 | *Ganoderma lucidum* (herb) | 2010 | Geographic origins | PCA<br>OPLS-DA | [1]H-NMR | [106] |
| 6 | *Panax ginseng, P. quinquefolius* (roots) | 2009 | Taxonomic discrimination | PCA | [1]H-NMR | [107] |
| 7 | *Arabidopsis* | 2009 | Taxonomic discrimination | PLS-DA | [1]H NMR | [108] |
| 8 | Green Tea | 2007 | Quality assessment | PCA<br>PLS<br>OSC | [1]H-NMR | [109] |

**Table 1.** *Cont.*

| No. | Plant Name | Publication Year | Research Aims | Applied Statistical Methods (If Any) | Analytical Techniques | References |
|-----|-----------|------------------|---------------|--------------------------------------|------------------------|------------|
| 9 | *Panax ginseng* Panax ginseng C.A. (roots) | 2007 | Quality assessment | PCA | [1]H-NMR | [110] |
| 10 | Brassica rapa (leaves) | 2007 | Geographic origins | PCA | [1]H-NMR | [111] |
| 11 | Ilex ssp. (arbutin) | 2005 | Discrimination | PCA | [1]H-NMR | [112] |
| 12 | Arabidopsis thaliana (seeds) | 2003 | Discrimination | PCA | [1]H-NMR | [113] |
| 13 | *Strychnos nux-vomica* (seeds, stem bark, root bark), *S. icaja* (seeds), *S. ignatii* (leaves, stem bark, root bark and collar bark) | 2004 | Discrimination | PCA | [1]H-NMR | [114] |
| 14 | *Artemisia annua* (herb) | 2004 | Geographic origins | PCA PLS PLS-DA | [1]H-NMR | [115] |
| 15 | *Camellia sinensis* (green tea leaves) | 2004 | Geographic origins | PCA HCA | [1]H-NMR | [116] |
| 16 | *Cannabis sativa* (flowers) | 2004 | Geographic origins | PCA | [1]H-NMR | [117] |
| 17 | Coffee | 2002 | Quality assessment | PCA LDA | [1]H-NMR | [118] |
| 18 | *Quillaja saponaria* (bark/saponins) | 2001 | Structural elucidation | PCA PLS-DA | [1]H-NMR | [119] |
| 19 | Propolis | 2016 | Classification | PLS-DA RF | | [120] |
| 20 | Ginseng Radix | 2010 | Evaluation | PCA CA | [1]H–NMR | [121] |
| 21 | *Isatis tinctoria* | 2015 | Comparison | PCA CA k-NN | [1]H–NMR | [122] |
| 22 | *genus Paeonia* L. | 2017 | Determination | PCA | [1]H–NMR | [123] |
| 23 | *Serenoa repens* | 2018 | Authentication of saw palmetto | PCA | [1]H–NMR | [124] |
| 24 | Polygoni Multiflori Radix | 2018 | Metabolomics | PLS-DA N3 | [1]H–NMR | [125] |
| 25 | Two cinnamon species | 2018 | Authentication | PCA OPLS-DA | [1]H–NMR | [126] |
| 26 | *Neptunia oleracea* | 2016 | Metabolites | PLSR | [1]H–NMR | [127] |
| 27 | *Orthosiphon stamineus* | 2011 | Metabolomics | PLSR OPLSR | [1]H–NMR | [128] |
| 28 | Saffron | 2010 | Discrimination | PCA | [1]H–NMR | [129] |
| 29 | *Angelica gigas* | 2011 | Metabolomics | PCA OPLS-DA | [1]H–NMR | [130] |

## 9. NMR-Based Metabolomics: What the Future Holds

### 9.1. Perspectives on the Economy

The food authentication, functional genomics, and the significant equivalence testing of GMOs are just a few prospective uses for NMR-based metabolomics. NMR, being a dependable, non-destructive method for QC of economically relevant samples, is robust, reliable, and non-destructive. The combination of NMR spectroscopy and multivariate statistical analysis software opens up new possibilities for conducting sound and accurate quality control analyses of botanical samples. The necessary standardization of HMs will most likely play a major part in the approval of much more complicated HMs as a therapy in the future, necessitating the robust and quick analytical methods for QC of these products without the time-consuming preparation required for QC on the complex mixtures. As a result, QC studies will most likely no longer be limited to a few chosen elements, but will instead cover the whole sample composition.

NMR metabolomics, when used in conjunction with functional genomics, can help researchers to gain a better understanding of the complex structure of the plant networks, and answer how they evolve as a result of genetic modification. Except for genetic modifications, the basic nature of plant phenotypes in connection with their development, physiology, and environment can also be established. It has been possible to employ metabolomics to determine the impact of environmental stress on root and particular enzyme activity [130]. It is also becoming increasingly common to combine the diverse "omics" datasets and cross-correlate them to extract as much information as possible from these data-rich matrices [88].

NMR-based metabolomics will play an important role in the future of many biological domains, both in terms of their research and economic growth. In the realm of drug discovery and development, NMR-based metabolomics will help to identify the leading components quickly and efficiently, and in certain situations, it may be able to replace time-consuming chromatography methodology [7].

*9.2. Perspective on Future Developments*

NMR technologies have recently advanced to the point where they are an acceptable approach possessing the specific characteristics for the investigation of plant metabolites. The resolution and sensitivity capabilities of NMR are its biggest flaws. MS has the extra advantage of being easily connected to a chromatography unit for combinational analysis, whereas NMR requires more effort to link with chromatography procedures. Chemical compounds can be recognized at very low concentrations, and even at the trace element level, using MS analysis, enriching the data for chemometric analysis. The complex plant extracts used in NMR create a lot of overlap in most regions of the spectrum, making it challenging to retrieve the information needed from the spectra.

Magical angle spinning (MAS)-NMR spectroscopy is another advancement that has received attention. This approach uses an extremely low quantity of deuterated solvent and has a very simple sample preparation process (e.g., direct insertions of lyophilized tissue on an MAS 4 mm zirconium rotor), resulting in the combined benefit of no sample preparation issues and minimal amounts of costly solvents used. The samples are then analyzed at a 54.7° angle (the magic angle), which considerably lowers line broadening and yields high-resolution spectra. This has the advantage of preventing the samples from being tainted by chemical reactions during their preparation [131,132].

## 10. Conclusions

With the public's growing concern about the protection and quality of HMs, the conforming ideals that efficiently analyze HMs created from raw materials to medical goods are becoming increasingly important for human health, as well as the industry's long-term viability. The chemical fingerprint utilized to provide a thorough description of complex matrices is promising, and is projected to become a potent tool for HM quality assessment due to the advantages of the overall quality assessment. The framework used with the chemical fingerprint and chemometric approaches (RMN) was presented and discussed in depth in this review. The utilization of chemometric approaches for data processing and well-established chemical fingerprint procedures for HM quality evaluation were considered. In conclusion, according to numerous research, a proper fingerprint analysis should comprise acceptable analytical methodologies, data pre-processing processes, and appropriate chemometric approaches, depending on the scope and purpose of the application. In the future, more studies will be needed to integrate chemical fingerprint-based chemometric analysis to the traditional quality evaluation of HMs. Hopefully, in the future, the growth of the chemical fingerprint paired with chemometric methodologies will be widened in terms of suitable fields for the quality surveillance of HMs, based on the standardization and validation of the entire framework.

Taking all of the foregoing into account, NMR-based metabolomics has a distinct advantage over other traditional approaches. This is because it enables one to analyze all of the molecules in the sample, providing a comprehensive picture of this sample. It also has the ability to distinguish between the samples of various sources.

**Author Contributions:** Conceptualization, A.R., B.B.S., H.H., S.Z., I.B.A., I.K. and M.M.; methodology, A.R., B.B.S., H.H., S.Z., I.B.A., I.K., M.M., A.P., G.C., S.S., M.A. and P.P.; validation, A.R., B.B.S., A.R., B.B.S., H.H., S.Z. and I.B.A.; writing—original draft preparation, M.A. and P.P.; writing—review and editing. All authors have read and agreed to the published version of the manuscript.

**Funding:** This research received no external funding.

**Institutional Review Board Statement:** Not applicable.

**Informed Consent Statement:** Not applicable.

**Data Availability Statement:** The data presented in this study are available on request from the corresponding author.

**Conflicts of Interest:** The authors declare no conflict of interest.

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
