# Peer review of "Quality Assessment of Medicinal Plants via Chemometric Exploration of Quantitative NMR Data: A Review"

_compounds, doi:10.3390/compounds2020012_

Round 1
Reviewer 1 Report
The review by Rebiai et al. summarizes a recent progress in the field of NMR-based metabolomics with a focus on herbal medicine analysis. While there are numbers of reviews of using NMR and chemometrics to study various plants-based extracts, this is the first one that specifically address the herbal medicine. The review is very detailed and covers the most recent papers in this field so it will be very helpful for specialists in NMR and chemometrics and interesting for the general readership of “Compounds” journal.
The review is very well-written, provides a nice overview of chemometrics methods for those who are new to the field. It might only require several minor corrections before it can be accepted:
- The most hard-to-grasp part of the review for most chemist will be the section 6, where the main ideas of chemometric approaches are described. While it is quite clear, given the complexity of the topic, it will benefit from some infographics that illustrate basic concepts of PCA and PLS as the main two representatives of data exploration approaches. Current Fig. 2, unfortunately, now do not help the understanding much.
- It is not clear for me why this review contains a section on Fruit juice QC (8.1), it has little to do with herbal medicine topic.
- There is a small number of typos that must be corrected (i.e. p.9 line 407).
Author Response
Replies are uploaded as a doc file.

Reviewer 2 Report
According to the article title, the article discusses a utilization of the NMR data for quality assessment of medical plants. It is a review article. The article is of possible interest for the NMR community or for the community of scientists focusing on qualitative analysis of preparations from medical plants. However, this is only the statement. The main text doesn’t bring anything new, especially its second half. At several points, I see an incomprehension of the basic principles of NMR-omics sciences, this has to be significantly improved. Starting with the chapter 4, the principles of compound profiling, fingerprinting, targeted analysis, qualitative and quantitative analysis are not described in the right way. Figure 1 misses important software used in metabolomics, chemometrics and multivariate data analysis is switched in the picture and so on. Row 216 is very confusing. Both figures contain typos. Chapter 7.2, the usual number of bins is several hundred not thousands. In profiling using NMR, usually less than 100 compounds is used and it is sufficient for PCA o OLPS-DA. Foodomics is just mentioned, there are hundreds of paper dealing with this topic (chapter 8.1). Also 13C NMR data are commonly used in foodomics, but there is no mention about that. I do not see any reason to have in such a general article table 1, which I firmly believe does not represent a representative overview. I do not understand why the two examples of herbal medicine analysis are present in detail in this rather general article (chapter 8.3 and 8.5). Perspective on future development seem more like a description of the MAS-NMR principles. I believe, that MAS-NMR has no future in herbal medicine analysis. Conclusion paragraph is also very weak.
Author Response
Replies are uploaded as a doc file.

Reviewer 3 Report
The authors present a nice review of NMR characterization of medicinal plants via chemometrics methods, with citation of many valuable examples.
The paper would benefit from adding a short section on data normalization, since this is a critical task in preparing NMR data for chemometric analysis.
There are some recommended minor language adjustments which follow, and the language of the paper would benefit from removing the word "the" from many places where it is used.
29 Since ancient times, herbal medicines (HM) have played such a vital role in worldwide healthcare systems that it is critical to conduct thorough quality evaluation and control of their complicated chemical makeup in order to ensure efficacy and safety.
33 “attractive” instead of “persuasive”
35 extremely valuable for inductive studies … as well as for solving industrial challenges
51 Statistically based algorithms such as principal 51 component analysis and free component examination can be used to separate spectral components without having comprehensive knowledge of the source signals from these components.
54 These are useful approaches, but due to implementation concerns such as software challenges, a lack of understanding, and a lack of literature examples, they are not as widely employed as they might be.
57 in NMR data analysis
58 It also describes the challenges in preparing NMR data for multivariate data processing
63 still used as a primary health tool in a number of developing
65 for basic healthcare
66 indiginous peoples residing in rural villages
72 gaining prominence
75 is not officially recognized
Therefore, support for education and research in this area has been lacking.
84 must be established
97 may have enhanced its higher
99 The inherent complexity of herbal therapy may be a critical aspect of its efficacy, and science is
127 contribute to the superiority
138 data to
139 The strength of chemometrics is that it can condense multidimensional data to a form that can be presented graphically to highlight the nature of similarities, differences, and trends.
143 Metabolomics is the
222 definitely has value, it has become a barrier
226 The majority of chemometrics methods work with PCs
227 Among its chief advantages is the capacity of PC models to manage
410 have been developed
Author Response
Replies uploaded as a doc file.
